# Investigation of the Adverse Events Associated with Bee Venom Pharmacopuncture in Patients Hospitalized in a Korean Hospital: A Retrospective Chart Review Study

**DOI:** 10.3390/toxins14100662

**Published:** 2022-09-23

**Authors:** In-Hu Bae, Woo-Sang Jung, Seungwon Kwon, Han-Gyul Lee, Seung-Yeon Cho, Seong-Uk Park, Sang-Kwan Moon, Jung-Mi Park, Chang-Nam Ko, Ki-Ho Cho

**Affiliations:** 1Department of Korean Medicine Cardiology and Neurology, Graduate School, Kyung Hee University, Seoul 02447, Korea; 2Department of Cardiology and Neurology, College of Korean Medicine, Kyung Hee University, Kyung Hee University Medical Center, Seoul 02447, Korea; 3Department of Cardiology and Neurology, College of Korean Medicine, Kyung Hee University, Kyung University Hospital at Gangdong, Seoul 02447, Korea

**Keywords:** bee venom pharmacopuncture, adverse event incidence rate, retrospective study

## Abstract

In bee venom pharmacopuncture (BVP), bee venom isolated from the venom sac of bees is injected into the acupoint or muscle associated with a disease. However, the histamine component in bee venom can cause adverse events; therefore, attention is required for BVP use. This study investigated the frequency, severity and characteristics of patients developing BVP-associated adverse events. The medical records of patients treated with BVP at Kyung Hee University Korean Medicine Hospital between 1 January 2013 and 1 May 2021 were reviewed. The demographic characteristics, disease-related characteristics, treatment-related characteristics and impressions of each patient were analyzed. In this study, >50% of 4821 inpatients were hospitalized for neurological disorders. The mean age of the overall study population was 54.62 ± 16.38 years and 61% were women. The frequency of adverse events was 2.32%. The mean age in the adverse events group was 58.20 ± 16.10 years and 76% were women. Two patients experienced moderate adverse events, with no commonality between these events. Every patient recovered naturally with no sequelae. The results showed that BVP is a relatively safe therapeutic method. However, further studies are needed to determine the frequency of adverse events and identify the causality between baseline characteristics and adverse events.

## 1. Introduction

Bee venom pharmacopuncture (BVP) is a treatment method used in traditional East Asian medicine. The method involves the injection of venom extracted and processed from the venom sac of honeybees into an acupoint or intramuscular pressure point associated with disease for therapeutic purposes [1,2,3]. Although BVP is used for the treatment of musculoskeletal disorders as well as various other diseases, some components in the body act as antigens and can cause life-threatening adverse events (AEs), such as anaphylaxis [4,5].

Therefore, skin testing is performed by subcutaneously injecting a solution containing bee venom (BV) before performing BVP to prevent AEs in the clinical setting. However, studies on causality between factors affecting BVP, such as dosage, concentration, purification, patient underlying diseases and AEs are needed to more effectively prevent AEs. In particular, the dilution of BV to a certain concentration and before injection for therapeutic purposes differs from general bee stings. Thus, the causality between BVP and AEs requires continuous evaluation separate from previous studies on allergic reactions to hymenoptera venom.

A single-center, large-scale retrospective study and systematic review of the incidence of AEs related to BVP as well as AE conditions has been conducted. However, since most previous studies included patients with musculoskeletal disorders, neoplasms or autoimmune diseases, there is a lack of studies on patients with internal diseases such as cardiovascular and respiratory disorders, who are relatively vulnerable to bee stings and patients with various disorders [6,7,8].

Therefore, the present study investigated the incidence of post-BVP AEs (including severe AEs such as anaphylaxis) by retrospective chart review and clinical characteristics and risk factors of patients with various disorders who were admitted to Kyung Hee University Korean Medicine Hospital complaining of AEs.

## 2. Results

From 1 January 2013 to 1 May 2021, a total of 5505 patients were treated with BVP. Among them, 684 patients were excluded since their medical records did not clearly describe BVP, such as missing data on patient condition after the treatment in the hospitalization medical records. Thus, this study included a total of 4821 patients (Figure 1).

### 2.1. Characteristics of Patients Administered BVP

Of 4821 patients that received BVP, 2945 (61%) were women, while 1876 (39%) were men. The mean age was 54.62 ± 16.38 years. A total of 62,413 treatments were administered, with a mean of 11.18 ± 8.49 treatments per patient. Patients were classified as eleventh revision of the International Classification of Diseases (ICD-11) [9]. The most common chief symptom was related to diseases of the nervous system (8A00–8E7Z) reported in ≥50% of the patients, followed by symptoms related to diseases of the musculoskeletal system or connective tissue (FA00–FC0Z). Of the total number of BVP treatments performed (62,413), 97% (60,754) was delivered at a concentration of 1:30,000 (Table 1).

### 2.2. Incidence of AEs after BVP

A total of 112 patients (86 women and 26 men) complained of AEs after receiving a total of 324 procedures. The incidence rates were 2.32% of the total of 4821 patients and 0.51% of 62,413 treatments. Of the 112 patients, two complained of systemic edema and shortness of breath, respectively, which were classified as moderate adverse events (2 of 62,413 treatments). Thus, the incidence of moderate AEs was 0.04% of total patients and 0.003% of total treatments. Most AEs occurred when BVP was delivered at a concentration of 1:30,000. The patients complained of the following AEs: itching (92 patients), local redness (73 patients), local edema (28 patients), subcutaneous bleeding (6 patients), vesicle (2 patients), purulent discharge (2 patients), local pain (1 patient), rash (1 patient), chest discomfort (1 patient) and systemic edema (1 patient). All symptoms resolved in a range of 24 h (1 day) to 552 h (23 days) (Table 2). Detailed information about the patients who exhibited AEs is provided in Appendix A (Table A1).

### 2.3. Comparisons of the Characteristics between Patients with and without Post-BVP AEs

The proportion of male patients in the group with AEs was significantly lower than that in the group without AEs (23.21% vs. 39.3%, *p* < 0.001). The mean age of patients with AEs was 58.20 ± 16.10 years, which was significantly higher than that of patients without AEs (54.53 ± 16.38 years). Regarding disease types, the proportion of AEs was relatively high in diseases of the ear or mastoid process (11.1%) and diseases of the musculoskeletal system (6.47%) (Table 3).

### 2.4. Characteristics of Patients Who Complained of Moderate AEs

Both patients who showed moderate AEs were women in their 50s or older. Both patients complained of AEs after receiving one BVP treatment. Patient 1 reported chest discomfort and itching, while Patient 2 reported systemic edema and itching. However, they did not experience life-threatening symptoms and the related symptoms completely disappeared within 24 h without particular treatment. Neither patient complained of resultant sequelae. A review of their medical records to investigate their prior medical history, treatment site, BVP concentration and hematology tests performed at the time of initial hospitalization, showed no common factors other than the treatment concentration used (Table 4).

## 3. Discussion

In traditional East Asian medicine, BVP is used for the treatment of various diseases. Studies on the incidence of AEs and relevant factors are continuously published to prevent and identify AEs that can occur when performing BVP [6,7,8,10]. 

The present study investigated the incidence of AEs in patients treated with BVP while hospitalized at Kyung Hee University Korean Medicine Hospital between 1 January 2013 and 1 May 2021. Among 4821 total patients, 112 patients reported 324 AEs. This accounts for 2.32% of the total number of patients and 0.51% of the total number of treatments administered (62,413). Of the 112 patients, two patients reported one moderate AE, respectively. Of them, Patient 1 complained of chest discomfort, while Patient 2 complained of systemic edema. However, none of the 112 patients showed anaphylactic shock. Patients fully recovered from AEs within a minimum of 24 h (1 day) or a maximum of 552 h (23 days), without other sequelae. Given these results, BVP can be considered a relatively safe treatment.

There was no occurrence of anaphylaxis in this study, which is lower than the result of previous study. Ko et al., reported that incidence rate of anaphylaxis was 0.045% in a systematic review [5]. However, the incidence of overall AEs in this study was higher than that reported by other retrospective studies investigating post–BVP AEs (0.025% by Lee et al. [8] and 0.23% by Kim et al. [7]). This difference is assumed to result from the proportion of women (≥60%), a higher mean number of procedures in the present study compared to other studies and differences between investigators in determining causality between the drug and AEs due to the limitation of the retrospective study design.

In general, drug adverse reactions (ADRs) are affected by sex, age, underlying disease, concomitant medication and the dose and frequency of drug administration [11,12]. In terms of sex, women are reportedly relatively more vulnerable to ADRs compared to men [13,14]. In the present study, the proportion of women in the group that reported AEs was significantly higher than that in the group that did not report AEs (*p* < 0.001). Other studies reported high proportions of AEs in female patients [8], while others reported similar proportions between men and women [7] or only a 9% higher proportion of women [8]. In contrast, in the present study, the proportion of women was very high, at 60% or higher.

The patients in the present study also received more BVP treatments compared to the numbers reported in similar studies. Dose and frequency have been reported to affect the development of ADRs [11]. Lee et al. [8] reported a 0.025% incidence of AEs and a mean of seven treatments. In contrast, the patients in the present study received an average of 11.18 ± 8.49 treatments. The increased exposures to BV likely affected the incidence of AEs.

In addition, domestic and overseas studies conducted on AEs of BVP reported incidence rates of AEs ranging from 0% to 90.63%, although similar criteria to those in this study were used to determine the causality between the drug and AEs [15]. The discordant results might stem from BVP performed by various doctors according to site, different investigators, or differences in the determination of causality due to the limitations of the retrospective study design.

Comparison of the patient groups with and without AEs showed a significantly higher proportion of women in the group with AEs (*p* < 0.001). The mean age was also significantly higher (*p* = 0.030) than that in the group without AEs. Thus, women and aged people may be more vulnerable to the development of AEs following BVP therapy. As mentioned above, these results are assumed to reflect the fact that women are more vulnerable to ADRs compared to men and pharmacokinetic changes in the body with age increased the risk of ADR development [16].

The incidence of AEs by treated disease was higher in diseases of the ear and mastoid process and diseases of the musculoskeletal system and connective tissue compared to those of other diseases. BVP therapy for diseases of the musculoskeletal system and connective tissue is likely to be performed mostly on one site unlike procedures performed for other diseases (for example, therapy is intensively performed on the lower back for patients with lumbar pain) [17,18]. These treatment patterns likely increase the incidence of AEs. However, regarding diseases of the ear and mastoid, it is difficult to discuss the clinical significance of these findings since few patients in this study received BVP for this condition.

The present study has some strengths. First, unlike previous studies, the present study assessed the safety of BVP in patients with chief complaints of various diseases [6,7,8]. As mentioned above, underlying diseases affect the development of ADRs [11,12]; however, previous studies on AEs from BVP included patients mainly presenting with chief complaints of diseases of the musculoskeletal system, injury, poisoning and certain other consequences of external causes for investigation, neoplasm or autoimmune system [6,7,8]. Meanwhile, the present study evaluated the safety related to BVP in patients hospitalized with chief complaints of diseases of the nervous or internal disease like endocrine, respiratory or circulatory systems, which no previous study has evaluated. The results of this study showed that the incidence of AEs was not significantly high in these patient groups.

Second, the results of the comparisons of the characteristics of the group of patients with and without AEs resulting from BVP confirmed that AEs resulting from BVP were affected by patient sex and age (women) similarly to AEs of other drugs. Furthermore, the incidence of AEs tended to be high in diseases of the musculoskeletal system, where BVP was intensively performed on one area, thus providing a useful clue for the application of BVP in clinical settings.

Lastly, the present study comprehensively investigated the characteristics of patients requiring BVP therapy, the number of treatments and the treatment concentrations in Korea to provide the beginning of the provision for guidelines for the administration of BVP.

However, the present study has some limitations. First, since most patients included in this study were older, patients of all ages (between their 20s and 50s) were not included. Moreover, most of the patients were female, which likely biased the results. Second, although the incidence of AEs by disease was compared, the analysis was conducted only using general disease classification according to the ICD-11 [9], instead of each disease. The number of patients for each group was not consistent and group sizes varied. These differences affected the accuracy of the results. Third, since the number of patients with moderate AEs was lower than those with mild AEs, the comparisons were not significant. It was difficult to find common ground between the patients in the group with moderate or greater AEs. Finally, the comparison of the incidence of AEs by BVP concentration was not significant because most patients were administered BVP at 1:30,000.

The results of this study demonstrated that BVP can be safely used for patients with various underlying diseases, including internal medicine and those of the nervous system, s well as aged patients. Although two patients showed moderate to greater AEs, these AEs were not life-threatening and the symptoms improved within 24 h without special measures and no sequelae. Thus, BVP therapy is relatively safe. However, since cases of moderate to greater AEs have been reported after multiple exposures to BVP [19], continuous monitoring is required if repeated BVP is needed. 

## 4. Materials and Methods

This study was conducted after receiving approval from the Institutional Review Board of Kyung Hee University Korean Medicine Hospital (KOMCIRB 2021–10–011). The study protocol was registered in the Clinical Research Information Service (CRIS) (KCT0006942).

### 4.1. Subjects

#### 4.1.1. Inclusion Criteria

The medical records of patients who met the following criteria were included in the analysis.

(1)Patients aged ≥18 years hospitalized at Kyung Hee University Korean Medicine Hospital and who received at least 1 BVP treatment between 1 January 2013 and 1 May 2021;(2)Patients with available medical records for identifying underlying diseases, demographic and clinical characteristics, BVP site and concentration and AEs occurring during the procedure.

#### 4.1.2. Exclusion Criteria

The medical records of patients who met the following criteria were excluded from the study.

If post-BVP AEs could not be identified because BVP site and concentration were not documented in the medical records during hospitalization.

### 4.2. Study Design

This retrospective medical record review study included data from the medical records of patients who received BVP therapy regardless of concentration and who met the above inclusion criteria. Information was collected from progress notes provided by the doctor of each patient, admission and discharge records, nursing progress notes and clinical observation notes.

### 4.3. Bee Venom 

The BV used for treatment was produced by the Korea Bee Venom Agriculture Corporation (Changnyeong, South Gyeongsang Province, Korea). Component analysis showed that melittin constituted 60% of the BV. Dried BV power was diluted in normal saline to concentrations of 1:1000–1:70,000. The diluted solution was injected intramuscularly or subcutaneously into an acupoint according to the disease at a minimum of one day apart or as needed.

### 4.4. Observation and Evaluation

Information about the following items was collected and evaluated by reviewing patient medical records.

#### 4.4.1. Characteristics of the Patients Administered BVP Therapy

Demographic characteristics and chief complaint (CC).

Patient medical records were reviewed to obtain data on demographic characteristics, such as sex, age and CC.

#### 4.4.2. BVP Concentrations

Patient medical records were reviewed to identify the concentrations administered for the treatments (1:1000–1: 70,000).

#### 4.4.3. Sites of BVP Therapy

Patient medical records were reviewed to determine the sites of BVP therapy: head (cephalus), face, neck, shoulders, waist, both upper extremities, both lower extremities, all extremities, left upper extremity, left lower extremity, right upper extremity and right lower extremity. In cases in which only the knees were involved, each was divided into left, right, or both knees.

### 4.5. Evaluation of Post-BVP AEs

#### 4.5.1. AE Incidence and Causality

After identifying the incidence of post-BVP AEs via patient medical records, the total number of patients who exhibited AEs and the number of treatments performed compared to the incidence of AEs were identified. Causality between BVP and AEs was evaluated using the World Health Organization-Uppsala Monitoring Centre (WHO-UMC) causality scale [20]. If the causality between BVP and an AE was “possible” or higher, the AE was considered to be caused by BVP.

#### 4.5.2. Evaluation of AE Severity

Patient medical records were reviewed to determine the AE severity. The severity was assessed using the three-category grading system described by Brown et al., in 2004 to evaluate anaphylaxis. The severity was categorized as mild, moderate, or severe depending on the changes in vital signs and onset site of AEs, as confirmed by hospitalization medical records [21]. The criteria for the three grades are described below.

-Mild: After BVP, an immediate or delayed reaction appearing at sites other than the treatment site, with symptoms confined to the skin and subcutaneous tissue-Moderate: Symptoms related to the respiratory, cardiovascular, or digestive systems in addition to symptoms confined to the skin and subcutaneous tissue-Severe: In addition to mild-moderate symptoms, confirmed changes in vital signs (i.e., hypoxia or hypotension) or neurological changes. Arterial oxygen saturation ≤ 92%, systolic blood pressure ≤ 90 mmHg, confusion, collapse, loss of consciousness, or incontinence.

#### 4.5.3. AE Onset Time

Patient medical records were reviewed to identify the number of treatments performed in patients who exhibited AEs from the time of the first treatment to AE onset. The unit was set to ‘number.’

#### 4.5.4. Time to Remission

Patient medical records were reviewed to identify the time elapsed from when patients who exhibited AEs first complained of AEs to when the symptoms completely disappeared, in hours.

### 4.6. Characteristics of Subjects with Moderate or Greater AEs

#### 4.6.1. Demographic Characteristics

Patient medical records were reviewed to identify demographic characteristics such as sex and age, in patients with moderate or greater AEs.

#### 4.6.2. Clinical Characteristics

Patient medical records were reviewed to identify clinical characteristics, such as moderate or greater underlying diseases in patients with moderate or greater AEs.

### 4.7. Statistical Analysis

IBM SPSS for Windows, version 25.0 (IBM Corp., Armonk, New York, NY, USA) was used to analyze the collected data. The statistical methods were as follows. Percentages were determined to verify the incidence of AEs. Means and standard deviation (SD) were used to assess patient sex and age, number of treatments, treatment site, AE onset time and number of complaints about AEs. Kolmogorov–Smirnov tests were used to test the normality of age of patient groups. Regarding age, chi-square tests were performed to compare and assess the significance of the differences in the number of treatments performed, age, or sex between each group. *p* < 0.05 was considered statistically significant.

## Figures and Tables

**Figure 1 toxins-14-00662-f001:**
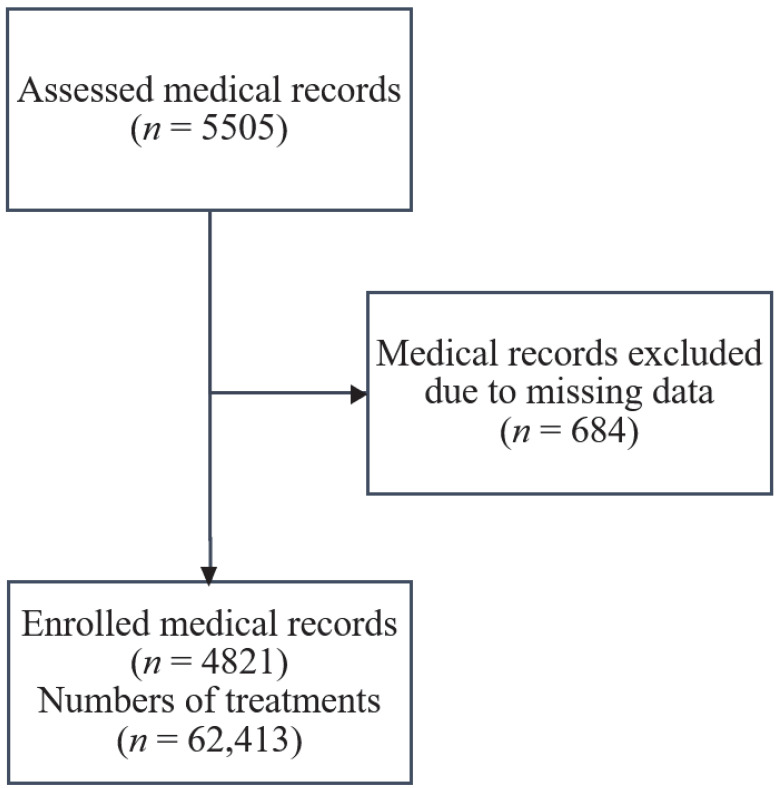
Study flow diagram.

**Table 1 toxins-14-00662-t001:** Demographic characteristics of the patients and specific details of bee venom pharmacopuncture (*n* = 4821, numbers of procedures = 62,413).

	Number (%) or Mean ± SD	Numbers of BVP Treatments per Patient
Age, years	54.62 ± 16.38	
Male	1876 (38.91)	
Disease Classification [9]		
Total numbers	4821 (100)	11.18 ± 8.49
1E50–1E5Z	2 (0.04)	8.5 ± 5.5
1F00–1F0Z	274 (5.68)	11.68 ± 6.16
2A00–2F9Z	56 (1.18)	10.91 ± 10.37
3A00–3C0Z	2 (0.04)	7.5 ± 3.5
5A00–5D46	9 (0.18)	7.4 ± 6.43
6A00–6E8Z	5 (0.10)	9 ± 5.76
8A00–8E7Z	2720 (56.44)	11.35 ± 6.79
9A00–9E1Z	11 (0.20)	9.36 ± 6.45
AA00–AC0Z	9 (0.18)	11 ± 11.93
BA00–BE2Z	390 (8.08)	12.62 ± 14.76
CA00–CB7Z	5 (0.10)	7.6 ± 4.67
DA00–DE2Z	28 (0.58)	8.14±6.92
EA00–EM0Z	6 (0.10)	9.8 ± 2.56
FA00–FC0Z	803 (16.65)	10.82 ± 9.24
GA00–GC8Z	31 (0.64)	12.00 ± 9.16
LA00–LD9Z	1 (0.02)	24
MA00–MH2Y	42 (0.87)	9.07 ± 9.35
NA00–NF2Z	421 (8.73)	9.66 ± 9.71
QA00–QF4Z	1 (0.02)	2
SD70-SD7Z	4 (0.06)	11.25 ± 7.82
SJ3Y–S3Z	1 (0.02)	9
Concentration of BVP treatments		
Total number of procedures	62,413 (100)	
1:10,000	156 (0.24)	
1:10,0000	25 (0.04)	
1:20,000	1221 (0.19)	
1:20,0000	4 (0.006)	
1:3000	411 (0.65)	
1:30,000	60574 (97.05)	
1:70,000	22 (0.03)	

SD: standard deviation; 1E50–1E5Z: viral hepatitis; 1F00–1F0Z: Viral infections characterized by skin or mucous membrane lesions; 2A00–2F9Z: Neoplasms; 3A00–3C0Z: Diseases of the blood or blood-forming organs; 5A00–5D46: Endocrine, nutritional or metabolic diseases; 6A00–6E8ZMental, behavioral or neurodevelopmental disorders; 8A00–8E7Z: Diseases of the nervous system; 9A00–9E1Z: Diseases of the visual system; AA00–AC0Z: Diseases of the ear or mastoid process; BA00–BE2Z: Diseases of the circulatory system; CA00–CB7Z: Diseases of the respiratory system DA00–DE2Z: Diseases of the digestive system; EA00–EM0Z: Diseases of the skin; FA00–FC0Z: Diseases of the musculoskeletal system or connective tissue; GA00–GC8Z: Diseases of the genitourinary system LA00–LD9Z: Developmental anomalies; MA00–MH2Y: Symptoms, signs or clinical findings, not elsewhere classified NA00–NF2Z Injury, poisoning or certain other consequences of external causes; QA00–QF4Z: Factors influencing health status or contact with health services; SD70–SD7: Supplementary Chapter Traditional Medicine Conditions—Module I, Qi, blood and fluid disorders; SJ3Y-S3Z: Supplementary Chapter Traditional Medicine Conditions—Four constitution medicine patterns.

**Table 2 toxins-14-00662-t002:** Summary of the characteristics of the patients who experienced adverse events (*n* = 4821, number of treatments = 62,413).

		Frequency (% of Total Patients)	Frequency (% of Total Treatments)
Total		112 (2.32)	324 (0.51)
Sex	Male (1876)	26 (1.38)	
Symptom severity			
	Grade 1 (mild)	110 (2.28)	324 (0.51)
	Grade 2 (moderate)	2 (0.04)	2 (0.003)
	Grade 3 (severe)	0 (0)	0 (0)
Concentration			
	1:3000	1 (0.02)	1 (0.001)
	1:10,000	1 (0.02)	2 (0.003)
	1:20,000	5 (0.10)	5 (0.008)
	1:30,000	105 (2.17)	316 (0.50)
Frequency of each adverse event symptom *			
	Itching	92 (1.90)	287 (88.58)
	Redness	73 (1.51)	204 (62.96)
	Local swelling	28 (0.58)	65 (20.06)
	Subcutaneous bleeding	6 (0.12)	9 (2.7)
	Vesicles	2 (0.04)	3 (0.92)
	Purulent	2 (0.04)	2 (0.61)
	Pain	1 (0.02)	1 (0.30)
	Rash	1 (0.02)	1 (0.30)
	Chest discomfort	1 (0.02)	1 (0.30)
	Systemic edema	1 (0.02)	1 (0.30)
Hours required for symptoms to disappear			Times (hours)
	Minimum		24
	Maximum		552
	Mean		63.42 ± 88.09

* If one patient complained of various symptoms, duplicate answers were made.

**Table 3 toxins-14-00662-t003:** Comparison of the characteristics of patients with and without adverse events after bee venom pharmacopuncture.

	Adverse Event Group (*n* = 112)	No Adverse Event Group (*n* = 4709)	*p* Value
Male, *n* (%)	26 (23.21)	1850 (39.3)	<0.001 *
Age, yrs, mean ± SD	58.20 ± 16.10	54.53 ± 16.38	0.030 **
Main impression of admission, *n* (%) [9]			
1E50-1E5Z (*n* = 2)	0 (0)	2 (100)	
1F00-1F0Z (*n* = 274)	0 (0)	274 (100)	
2A00–2F9Z (*n* = 56)	4 (7.02)	52 (92.98)	
3A00–3C0Z (*n* = 2)	0 (0)	2 (100)	
5A00–5D46 (*n* = 9)	0 (0)	9 (100)	
6A00–6E8Z (*n* = 5)	0 (0)	5 (100)	
8A00–8E7Z (*n* = 2720)	29 (1.06)	2691 (98.93)	
9A00–9E1Z (*n* = 11)	0 (0)	11 (100)	
AA00–AC0Z (*n* = 9)	1 (11.11)	8 (88.88)	
BA00–BE2Z (*n* = 390)	10 (2.56)	380 (97.43)	
CA00–CB7Z (*n* = 5)	0 (0)	5 (100)	
DA00–DE2Z (*n* = 28)	0 (0)	28 (100)	
EA00–EM0Z (*n* = 6)	0 (0)	6(100)	
FA00–FC0Z (*n* = 803)	52 (6.47)	751 (93.52)	
GA00–GC8Z (*n* = 31)	1 (3.22)	30 (96.77)	
LA00–LD9Z (*n* = 1)	0 (0)	1(100)	
MA00–MH2Y (*n* = 42)	0 (0)	42 (100)	
NA00–NF2Z (*n* = 421)	14 (3.32)	407 (96.67)	
QA00–QF4Z (*n* = 1)	0 (0)	1 (100)	
SD70-SD7Z (*n* = 4)	0 (0)	4 (100)	
SJ3Y-S3Z (*n* = 1)	0 (0)	1 (100)	

* *p*-value calculated by chi-square test, ** *p*-value calculated by Mann–Whitney U test. SD: standard deviation; 1E50–1E5Z: viral hepatitis; 1F00–1F0Z: Viral infections characterised by skin or mucous membrane lesions; 2A00–2F9Z: Neoplasms; 3A00–3C0Z: Diseases of the blood or blood-forming organs; 5A00–5D46: Endocrine, nutritional or metabolic diseases; 6A00–6E8ZMental, behavioural or neurodevelopmental disorders; 8A00–8E7Z: Diseases of the nervous system; 9A00–9E1Z: Diseases of the visual system; AA00–AC0Z: Diseases of the ear or mastoid process; BA00–BE2Z: Diseases of the circulatory system; CA00–CB7Z: Diseases of the respiratory system DA00–DE2Z: Diseases of the digestive system; EA00–EM0Z: Diseases of the skin; FA00–FC0Z: Diseases of the musculoskeletal system or connective tissue; GA00–GC8Z: Diseases of the genitourinary system LA00–LD9Z: Developmental anomalies; MA00–MH2Y: Symptoms, signs or clinical findings, not elsewhere classified NA00–NF2Z Injury, poisoning or certain other consequences of external causes; QA00–QF4Z: Factors influencing health status or contact with health services; SD70–SD7: Supplementary Chapter Traditional Medicine Conditions—Module I, Qi, blood and fluid disorders; SJ3Y-S3Z: Supplementary Chapter Traditional Medicine Conditions—Four constitution medicine patterns.

**Table 4 toxins-14-00662-t004:** Characteristics of patients with grade 2 adverse events.

	Unit	Patient 1	Patient 2
Age		95	51
Sex		Female	Female
Symptoms of adverse events		Itching Chest discomfort	Itching Systemic edema
Treatment site		Knee joint	Face
Concentration of bee venom pharmacopuncture		1:30,000	1:30,000
Past history		Gonarthrosis (knee) Hypertension Fibromyalgia Dyspepsia	Bell’s palsy Liver adenoma Gall bladder polyp Right renal cyst
WBC	10^3^/μL	6.73	13.18
RBC	10^6^/μL	3.86	4.73
Hemoglobin	g/dL	11.9	13.3
Hematocrit	%	36.2	39.7
MCV	fL	93.8	83.9
MCH	pg	30.9	28.1
MCHC	g/dL	33.0	33.5
Platelet count	10^3^/μL	304	229
MPV	fL	7.3	6.7
ESR	mm/hr	10	27
LUC	%	1.5	0.8
Segmented neutrophil	%	71.1	84.9
Lymphocyte	%	18.5	9.3
Monocyte	%	6.2	4.8
Eosinophil	%	2.1	0.1
Basophil	%	0.5	0.1
%Polymorpho-nuclear cell	%	74.7	82.7

WBC: white blood cell; RBC: red blood cell; MCV: mean cell volume; MCH: mean corpuscular hemoglobin; MCHC: mean corpuscular hemoglobin concentration; MPV: mean platelet volume; ESR: erythrocyte sedimentation rate; LUC: large unstained cell.

## Data Availability

The data presented in this study are available on request from the corresponding author. The data are not publicly available due to protection of personal information.

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
