# Peer review of "Investigation of the Adverse Events Associated with Bee Venom Pharmacopuncture in Patients Hospitalized in a Korean Hospital: A Retrospective Chart Review Study"

_toxins, 2022, doi:10.3390/toxins14100662_

Round 1
Reviewer 1 Report
This is an interesting retrospective study investigating the adverse events associated with bee venom pharmacopuncture in patients hospitalized in a Korean hospital.
Data have been properly analyzed and commented on. Furthermore, the presence of tables facilitates the presentation and understanding of the results.
Some suggested revisions are listed below and included in the manuscript.
Line 62: Add a reference to ICD in the legend and text. ICD is mentioned as an acronym but not explained appropriately. Korean classification is mentioned in line 211 (Discussion), but this is not the same subject.
Line 67: Please check the headings of table 1.
Table 1: Add a reference to ICD in the legend and text.
Line 95: Please explain what you mean.
Line 212: See comment about ICD in Table 1.
Line 327: Add a reference to ICD.

Reviewer 2 Report
I read the MS entitled "Investigation of the adverse events associated with bee venom pharmacopuncture in patients hospitalized in a Korean hospital: A retrospective chart review study" submitted for publication to Toxin.
The MS makes an exciting and valuable retrospective analysis of about 5000 patients who were treated from 2013 to 2021 with Bee Venom Pharmapuncture (BVP) at Kyung Hee University Korean Medicine Hospital. Two main aspects were investigated: 1) the incidence of adverse reactions and 2) the possible correlations of these last not only with musculoskeletal diseases (which represent a large proportion of the treated patients) but also with more general diseases like cardiovascular and respiratory diseases which could be more dangerous conditions for adverse effects of BVP.
The results seem to indicate that BVP is a relatively safe therapeutic method, even if some attention should be considered to suggesting further deeper analyses in specific types of patients.
The study is well-written and clear. Large Tables help the reader to understand the different aspects of the investigated patients.
Discussion is clear and in my opinion opened a sufficient criticism for further investigations.
Reviewer 3 Report
The manuscript is very interesting and brings important issue about bee venom acupuncture and its adverse effects. However, the study does not represent a novelty, as very similar articles have already been published. In this year, for example, one systematic review about bee venom acupuncture was published in Toxins (and by the way, it was not cited).
Although authors bring detailed information about adverse effects of bee venom acupuncture, the conclusion is that the incidence of such effects was not significantly high in patients with cardiovascular diseases, a group chosen for evaluation, already analyzed in previous studies. Moreover, other diseases not reported before, such as endocrine, were not discussed, probably because there are no consequences for the bee therapy. Thus, new results that the manuscript bring do not worth a new publication, unless that this novelty is well discussed and correlated to possible adverse symptoms.
Lastly, although there are few studies about this issue, a systematic review containing only 14 references is not adequate. More studies, especially in the discussion section, should be included to explain the results.
Round 2
Reviewer 3 Report
I understand that the manuscript is focused on the adverse effects, with data not published before, but I still think that the discussion should be better supported by scientific studies. Although the authors have added references, these new articles were not discussed, and it is not a systematic review, but important citations are still missing, such as Toxins 2022, 14(4), 238; https://doi.org/10.3390/toxins14040238, Toxins 2022, 14(1), 18; https://doi.org/10.3390/toxins14010018, Toxins 2021, 13(9), 608; https://doi.org/10.3390/toxins13090608, Toxins 2020, 12(9), 558; https://doi.org/10.3390/toxins12090558. These references may improve the discussion section and to compare results from different studies, since adverse effects and an indication of venom therapy is relevant.
